# Flavor Precursors and Volatile Compounds Improvement of Unfermented Cocoa Beans by Hydrolysis Using Bromelain

**DOI:** 10.3390/foods12040820

**Published:** 2023-02-14

**Authors:** Krisna Purbaningrum, Chusnul Hidayat, Lucia Dhiantika Witasari, Tyas Utami

**Affiliations:** Department of Food and Agricultural Product Technology, Faculty of Agricultural Technology, Gadjah Mada University, Yogyakarta 55281, Indonesia

**Keywords:** enzymatic hydrolysis, unfermented cocoa beans, free amino acids, cocoa flavors

## Abstract

Cocoa fermentation is an essential process that produces flavor precursors. However, many small farmers in Indonesia directly dry their cocoa beans without fermentation due to low yield and long fermentation time, resulting in fewer flavor precursors and cocoa flavor. Therefore, this study aimed to enhance the flavor precursors, particularly free amino acids and volatile compounds, of unfermented cocoa beans by hydrolysis, using bromelain. Unfermented cocoa beans were previously hydrolyzed with bromelain at concentrations of 3.5, 7, and 10.5 U/mL for 4, 6, and 8 h, respectively. An analysis of enzyme activity, degree of hydrolysis, free amino acids, reducing sugar, polyphenols, and volatile compounds was then conducted using unfermented and fermented cocoa beans as negative and positive controls, respectively. The results showed that the highest degree of hydrolysis was 42.95% at 10.5 U/mL for 6 h, although it was not significantly different from the hydrolysis at 3.5 U/mL for 8 h. This indicates a higher reducing sugar and lower polyphenols content than unfermented cocoa beans. There was also an increase in free amino acids, especially hydrophobic amino acids, such as phenylalanine, valine, leucine, alanine, and tyrosine, and desirable volatile compounds, such as pyrazines. Therefore, this suggests that hydrolysis with bromelain increased the flavor precursors and cocoa-bean flavors.

## 1. Introduction

The cocoa-bean flavor is an essential quality attribute that influences the acceptance of cocoa products [1]. The development of cocoa flavor is largely determined by the genetic profile of cocoa beans, the growing environment, and the processing methods used, such as fermentation and the drying process [2]. Fermentation produces volatile acidity of about 2% of dry matter. Acetic acid accounts for 90% of the total acids and plays an important role in catalyzing the enzymatic processes to develop flavor precursors [3]. During the fermentation process, endogenous peptidases break down proteins into small peptides and free amino acids, while invertase converts sucrose into reducing sugars. Additionally, the polyphenols decrease due to oxidation by polyphenol oxidase [1]. The main phenolic compounds in cocoa beans are flavan-3-ols (epicatechin and catechin), anthocyanins, and flavanols [4]. During roasting, Maillard reactions between free amino acids and reducing sugars as flavor precursors produce compounds such as pyrazines, aldehydes, and esters [5,6].

African countries such as Côte d’Ivoire, Ghana, and Cameroon are the major producers of cocoa beans, supplying 74.5% of the global production, with a total cocoa production of 3.6 million tons in 2021. Meanwhile, Asian countries supply 5.5% of global production, with Indonesia producing 200 thousand tons of cocoa in 2021 [7]. After Ivory Coast and Ghana, Indonesia is the third largest exporter of cocoa beans. About 88.48% of cocoa beans in Indonesia are managed by small farmers, and almost 80% of cocoa production is exported to the international market [8].

Many small Indonesian farmers dry fresh cocoa beans without fermentation due to low yield, long fermentation time, and a price difference that is not significant [9,10]. In addition, many small producers in other countries also dried their cocoa beans without the fermentation process. About 30% of small farmers in Ecuador sell cocoa beans without fermentation and drying processes [11]. The development of flavor precursors is critical during fermentation because fermented cocoa beans contain more flavor compounds than unfermented beans [5]. The quality of unfermented cocoa beans can be improved by fermentation, using external microbes; however, this process takes up to five days [12].

To produce free amino acids as one of the flavor precursors of unfermented dried cocoa beans, protein hydrolysis can be carried out using peptidases, which break down proteins into smaller peptides and free amino acids. Protein hydrolysis in fermented and dried cocoa beans, using exogenous peptidases (Flavourzyme) at 50 °C for 6 h, was reported to enhance the chocolate flavor. Therefore, the peptidases enzyme was useful in increasing cocoa flavor precursors and the flavor perception of cocoa products [13].

Enzymatic hydrolysis can be conducted using various peptidases, including bromelain, which is extracted from pineapple fruit (*Ananas comosus*). Bromelain is an endopeptidase with broad specificity for cleaving peptide bonds, particularly those with hydrophobic amino acid residues [14]. It is widely used due to its stable activity over a wide temperature, from 35.5 to 71 °C, and pH range from, 4.0 to 8.0 [14,15]. A previous study found that enzymatic hydrolysis using bromelain as endopeptidases in mung bean, brown rice, and seaweed (*G. fisheri*) by-products resulted in higher amounts of hydrophobic amino acids, which are significant in the formation of flavor compounds [14,16,17].

This study aimed to enhance the flavor precursors of unfermented cocoa beans, especially free amino acids, by enzymatic hydrolysis, using bromelain at various enzyme concentrations and times, to produce higher desired volatile compounds as cocoa-bean flavors.

## 2. Materials and Methods

### 2.1. Materials

Cocoa beans of the forastero variety with the characteristics of purple beans were collected from cocoa farmers in Patuk, Gunung Kidul, Yogyakarta, Indonesia. Crude bromelain was purchased from Chemic Lab. KP, Bogor, Indonesia, while casein from bovine milk, standard tyrosine, Na_2_CO_3_, Folin–Ciocâlteu reagent, hydrochloric acid, petroleum ether, and trichloroacetic acid were purchased from Merck (Darmstadt, Germany).

### 2.2. Preparation of Unfermented Dried Cocoa Beans

Cocoa pods were cleaved, and cocoa beans with shells were separated and then dried in a cabinet dryer at 50 °C for 24 h, until the moisture content was under 7.5%. Cocoa beans were then stored at room temperature before being used for analysis.

### 2.3. Bromelain Activity Assay

Enzyme activity assay was performed using a method by Cupp-Enyard and Aldrich (2008) [18] at various pHs, i.e., 5, 5.5, 6, 6.5, and 7, and temperatures of 40, 45, 50, 55, and 60 °C.

### 2.4. Enzymatic Hydrolysis of Unfermented Cocoa Beans Using Bromelain

Crude bromelain at concentrations of 3.5, 7, and 10.5 U/mL was added to a 0.05 M acetate buffer solution at pH 6. The solution was homogenized. Unfermented cocoa beans were added with a ratio of acetate buffer: cocoa beans 1:3 (*w*/*v*); incubated for 4, 6, and 8 h at 50 °C in a water-bath shaker (Memmert WNB14, Germany); and the degree of hydrolysis of cocoa beans was analyzed. Selected hydrolyzed cocoa beans were dried in a cabinet dryer at 65 °C for 24 h, and amino acids, reducing sugar, and total polyphenols were analyzed. Subsequently, dried cocoa beans were roasted at 140 °C for 30 min and then analyzed for volatile compounds, using SPME–GC–MS.

### 2.5. Degree of Hydrolysis Analysis

The degree of hydrolysis (DH) of the cocoa samples was analyzed according to a method by Hoyle and Merritt [19]. The Kjeldahl method was used to determine the total nitrogen and the nitrogen soluble in the supernatant. The degree of hydrolysis was calculated with the following equation:$$\mathrm{DH}\;(\%):\;\frac{Nitrogen\;Soluble\;in\;TCA\;20\%}{Total\;Nitrogen}\times 100$$

### 2.6. Soluble Protein Analysis

The soluble protein content in the buffer-enzyme solution was evaluated by using the Bradford method [20].

### 2.7. Free Amino Acid Analysis

The amino acid composition was analyzed according to the method described by Li [21] with modifications. Cocoa beans weighing 5 g were defatted in 25 mL of n-hexane solvent for 8 h, using a Soxhlet apparatus. A total of 60 mg of cocoa sample was added to 4 mL of 6 N HCl and then heated for 1 h at 110 °C. The solution was neutralized to pH 7 with 6N NaOH, then diluted to 10 mL, and filtered through a 0.2 μm Whatman filter paper. Samples of 50 μL were added to 300 μL of OPA solution before stirring for 5 min and injecting 10 μL into HPLC (Thermo Dionex Ultimate 3000, ThermoFisher Scientific, Waltham, MA, USA). The column used was LiChrospher 100 RP-18 (5 μm) with a Thermo Ultimate 3000 RS Fluorescence Detector. Mobile phase A was CH_3_OH: 50 mM sodium acetate: THF in a ratio of 2:96:2 and pH 6.8, while B was 65% CH_3_OH. The eluent gradient was 100% (A) in the range of 0.1–15 min, 35% (B) from 15 to 30 min, and 100% (B) from 30 to 40 min, and the flow rate was 1.5 mL/min.

### 2.8. Reducing Sugar Analysis

The Nelson–Somogyi method was used to determine reducing sugar content [22].

### 2.9. Total Polyphenols Analysis

Total polyphenols content was measured using the Folin-and-Ciocâlteu method [23].

### 2.10. Volatile Compound Analysis

The volatile aroma compounds were analyzed based on a modified method described by Caprioli [24]. In a 22 mL SPME vial, 1 g of cocoa-bean sample was placed and then heated in a water bath before extraction with 2 cm SPME fiber DVB/CAR/PDMS at 55 °C for 90 min. Furthermore, the sample was injected into the GC–MS instrument (Agilent 7890A, Agilent 5975C XL EI/CI, Santa Clara, CA, USA) with the injector temperature at 250 °C in splitless mode. The column used was DB-Wax (30 m × 250 µm × 0.25 µm), and the initial column temperature was 40 °C for 5 min and then increased by 3 °C/min to 220 °C. The flow rate (Helium) was 1 mL/min, and the identification of volatile components was carried out using the NIST 14 library.

### 2.11. Statistical Analysis

Data were analyzed using ANOVA (Analysis of Variance) at a significance level of 5% (*p* < 0.05), using SPSS 25.0 software. A further test with Duncan Multiple Range Test was used if there were significant differences in the results. Additionally, the mean values of the triplicate measurements were reported.

## 3. Results and Discussion

### 3.1. Bromelain Activity at Various pH Values and Temperatures

In this study, enzyme activity was measured to determine the pH and temperature of bromelain with the highest enzyme activity in the hydrolysis process.

The highest bromelain activity was found at 50 °C. As shown in Table 1, the bromelain activity increased with the increasing temperature up to 50 °C and then decreased. This is consistent with previous studies, which state that the activity increased with an increasing temperature from 30 °C to 50 °C and then decreased with a higher temperature. the highest enzyme activity of crude bromelain was at 50 °C, resulting in a high degree of hydrolysis [25,26]. According to Table 2, the highest enzyme activity was obtained at pH 6, as the activity increased with the increasing pH up to 6 and then decreased. It was previously reported that measurements of bromelain activity at pH 4.0 to 8.0 showed the highest activity at pH 6 [27], and a temperature of 50 °C and pH 6.0 were used in the hydrolysis process [28].

The results of this study showed that the bromelain enzyme conditions with the highest enzyme activity were at a temperature of 50 °C and pH 6.0, with an enzyme activity of 0.21 U/mL. Furthermore, enzymatic hydrolysis of unfermented dried cocoa beans was carried out under these conditions.

### 3.2. Hydrolysis of Unfermented Cocoa Beans Using Various Bromelain Concentrations and Time

Figure 1 shows the degree of hydrolysis of unfermented, fermented, and unfermented cocoa beans hydrolyzed by bromelain. The unfermented cocoa beans had the lowest degree of hydrolysis, while fermented cocoa beans had the highest, with values of 13.51% and 51.70%, respectively. The DH is assumed to only consist of short-chain peptides and amino acids as a result of protein hydrolysis [29]. Proteins are broken down into free amino acids as flavor precursors by endogenous peptidases during fermentation [1].

As a control, unfermented cocoa beans were incubated in an acetate buffer for 8 h, resulting in an increase in the DH from 13.51% to 23.30%. This increase could be explained by the action of endogenous peptidases, which react with proteins in cocoa beans and hydrolyze them into simpler compounds [13]. A higher accumulation of free amino acids was reported with incubation of unfermented cocoa beans in acetate buffer at 45 °C and pH 5.5. This is attributed to the carboxypeptidase and aspartate endopeptidases, which are endogenous peptidases in cocoa beans that could reactivate and degrade bean protein, resulting in higher free amino acids. However, the concentration of free amino acids did not reach the fermented cocoa beans until the incubation was complete at 16 h [30]. Another study reported that protein breakdown by endogenous peptidases occurred during the incubation of cocoa beans with acetic and lactic acids for up to 48 h [31].

The addition of exogenous peptidases (bromelain) significantly increased the DH of cocoa beans up to 76.98–83.07% of fermented cocoa beans. There was a rapid increase in DH during the initial 4 h of hydrolysis to 28.96%, 35.16%, and 39.70% for enzyme concentrations of 3.5 U/mL, 7.0 U/mL, and 10.5 U/mL, respectively. At low bromelain concentrations of 3.5 U/mL, the DH increased after 8 h. However, after 6 h of hydrolysis, there was no significant increase at higher enzyme concentrations of 7 and 10.5 U/mL. Furthermore, the DH of cocoa beans was almost similar at 39.05–40.97% after 8 h of incubation. In the hydrolysis of brown rice protein using bromelain, the DH value increased with enzyme concentration from 0 to 10% (*w*/*w*) and 0 to 3 h of hydrolysis, resulting in a value of 40.10% before plateauing. This could be due to the inhibitory effect of the end product [17].

Figure 1 shows that the highest DH was 42.95% at a bromelain concentration of 10.5 U/mL for 6 h of hydrolysis, but it was not significantly different from the value at a concentration of 3.5 U/mL for 8 h. The DH at a high concentration with a short hydrolysis time was the same as that at a low enzyme concentration with a long hydrolysis time [32]. In previous studies, the hydrolysis of mung beans with bromelain at a concentration of 20% (*w*/*w*) for 6 h produced the highest DH of 50.4% but was not significantly different from 15% (*w*/*w*) at 12, 18, and 24 h [16]. In the hydrolysis of seaweed protein by-products, using bromelain resulted in an increase in the DH and reached a plateau after 6 h, with a DH of 62.91%. However, DH values of 15% (*w*/*w*) and 20% (*w*/*w*) were not significantly different [14].

Based on Figure 1, the DH did not increase linearly with the increasing enzyme concentration. This was possibly due to the limitation of enzyme diffusion into the beans. For protein hydrolysis to occur, bromelain must diffuse into the beans and come into contact with the cocoa protein. The inhibition of enzyme diffusion will lead to a decrease in enzyme reaction rate [32]. This is consistent with the results of the soluble protein analysis in the buffer–enzyme solution. Figure 2 shows that the soluble protein was significantly reduced (*p* < 0.05) after 2 h of hydrolysis. This indicates that the enzyme can diffuse into the beans and play a role in hydrolysis. However, a 2-fold increase in bromelain concentration did not result in a 2-fold decrease in soluble protein content in the buffer and was relatively constant for up to 8 h of hydrolysis. It appeared that enzyme diffusion into the beans is limited, so the hydrolysis of protein in the seed is influenced not only by enzyme activity but also by the contact between the enzyme and the protein in the beans.

The DH in cocoa beans hydrolyzed by bromelain was significantly higher than in unfermented cocoa beans but lower than in fermented cocoa beans. A DH of 40% is acceptable, and a value of 50% indicates that the beans are well fermented [33]. Therefore, based on the highest DH and the DH above 40%, bromelain concentrations of 10.5 U/mL for 6 h and 3.5 U/mL for 8 h of hydrolysis were used for further analysis of free amino acids, reducing sugar, total polyphenols, and volatile compounds.

### 3.3. Formation of Flavor Precursor in Unfermented Cocoa Beans with the Addition of Bromelain

#### 3.3.1. Free Amino Acids

Free amino acids and reducing sugars are flavor precursors in cocoa beans which develop into a cocoa flavor during roasting by the Maillard reaction [34]. Table 3 shows that free amino acids in fermented cocoa beans were higher than in unfermented cocoa beans. The increase in free amino acids in fermented cocoa beans was due to the activity of endogenous peptidases. During fermentation, microbial activity produces heat and acetic acid, which diffuse into the beans, causing the death of the cocoa beans. Furthermore, endogenous peptidases are activated and react with bean proteins to produce amino acids as flavor precursors [35]. The primary precursors that contribute to flavor development are hydrophobic amino acids formed during fermentation, particularly alanine, leucine, isoleucine, phenylalanine, and valine [36,37].

The free amino acids after enzymatic hydrolysis were higher than in unfermented cocoa beans and were almost similar to those of fermented cocoa beans, as shown in Table 3. Enzymatic hydrolysis increased hydrophobic amino acids, such as alanine, tyrosine, valine, leucine, phenylalanine, and glycine, as the main precursors and other amino acids due to protein breakdown by bromelain during hydrolysis. Bromelain has a broad specificity for protein breakdown of both polar amino acids and hydrophobic amino acids [38]. Cleavage by bromelain is mainly in hydrophobic amino acid residues, while non-polar amino acid residues remain at the C-terminus of the peptide. Bromelain has a broad specificity for protein breakdown, especially in hydrophobic amino acid residues, while non-polar amino acid residues remain at the C-terminus of the peptide. Its specific cleavage is at the arginine–alanine and alanine–glutamic acid bonds [14,17] in the order of lysine, glutamic acid, glycine, and alanine [39]. Enzymatic hydrolysis using bromelain increases arginine, alanine, glutamic acid, and asparagine [38].

Free amino acids at enzymatic hydrolysis treatment with a bromelain concentration of 3.5 U/mL for 8 h were higher than 10.5 U/mL for 6 h. However, both were increased compared to unfermented cocoa beans, suggesting that enzymatic hydrolysis with bromelain could increase free amino acids as flavor precursors, especially hydrophobic amino acids, which primarily contribute to the development of the cocoa-bean flavor.

#### 3.3.2. Reducing Sugar

Fermented cocoa beans had a higher reducing sugar content (11.85%) compared to unfermented cocoa beans (4.60%), as shown in Figure 3. During fermentation, invertase breaks down sucrose into glucose and fructose as flavor precursors [40]. Subsequently, the sucrose concentration decreases, while glucose and fructose, as reducing sugars, increase [41]. In a previous study, the reducing sugar increased from 4.5%–4.69% to 10.5% after fermentation [42].

The reducing sugar in cocoa beans with enzymatic hydrolysis was significantly higher (7.34–8.16%) than in unfermented cocoa beans; this increase was 61.94 to 68.86% of fermented cocoa beans. Since the bromelain enzyme specifically cleaves proteins [14,17], the increase in reducing sugars was probably caused by incubation in acetate buffer at 50 °C and pH 6, which could activate endogenous invertase. This is consistent with a previous study, which found that incubating dried unfermented cocoa beans in acetate buffer for 16 h increased the concentration of reducing sugars (fructose and glucose). This indicated that the incubation treatment activated the remaining invertase to convert sucrose into reducing sugars [30]. Furthermore, incubation in acetic acid at pH 4–5 and 40–50 °C decreased the sucrose concentration, while the glucose and fructose concentrations increased [43].

### 3.4. Polyphenols Content in Unfermented Cocoa Beans Hydrolyzed by Bromelain

Figure 4 shows that total polyphenols content was significantly lower in fermented (11.70 mg GAE/g) compared to unfermented cocoa beans (25.08 mg GAE/g). The decrease in the polyphenols content is due to its reduction by polyphenol oxidase during the fermentation and drying process [44]. In a previous study, the total polyphenols content of fermented cocoa beans was 10.53–12.96 mg/g [45].

The total polyphenols content of cocoa beans by enzymatic hydrolysis was 20.80–19.75 mg GAE/g, which was significantly lower than that of unfermented and higher than that of fermented cocoa beans. However, this decrease was only 31.9–39.9% of fermented cocoa beans, as shown in Figure 4. Due to the specificity of bromelain for cleavage protein to produce amino acids [14,17], enzymatic hydrolysis using bromelain reduced the total polyphenols. This decrease could be attributed to the activation of endogenous polyphenol oxidase due to the incubation in acetate buffer at 50 °C and pH 6. In previous studies, the incubation of dried unfermented cocoa beans in acetate buffer resulted in the activation of endogenous enzymes in the cocoa beans. The incubation of cocoa beans may have activated polyphenol oxidase to oxidize polyphenols due to suitable conditions, such as the temperature and the presence of water [46].

A decrease in flavor formation and an increase in astringency and bitterness were associated with an increase in polyphenols’ concentration [5]. High concentrations of polyphenols negatively impact cocoa flavor. Furthermore, the concentrations of amino acids and reducing sugars as flavor precursors and flavors formed during roasting decreased with higher concentrations of polyphenols. This is due to the strong tendency of polyphenols to bind to other compounds, such as proteins and sugars [34,47].

### 3.5. Volatile Compounds of Unfermented Cocoa Beans Hydrolyzed by Bromelain

Fermentation and enzymatic hydrolysis increased the desirable volatile compounds in unfermented cocoa beans, such as pyrazines, aldehyde, and esters, as shown in Table 4 [6]. The increase in desirable volatile compounds is due to an increase in free amino acids and reducing sugars in flavor precursors. Additionally, cocoa beans with higher-flavor compounds were possibly due to the higher-flavor precursors produced during fermentation [48]. The Maillard reaction is based on the reaction between reducing sugars and free amino acids as flavor precursors. The initial reaction occurs between the flavor precursors to produce Amadori compounds via the Amadori rearrangement. Further reactions, such as Strecker degradation, form volatile compounds with characteristic cocoa flavor [40,49]. A previous study found that volatile compounds in roasted cocoa beans are pyrazines (21.67%), acids (26.71%), aldehydes (12.61%), alcohols (12.67%), ketones (5.97%), and esters (2.46%) [50].

Table 4 shows that the percentage of total pyrazine was higher in enzymatic hydrolysis than in unfermented cocoa but lower in fermented cocoa beans. Tetramethyl-pyrazine, trimethyl-pyrazine, 2,3-dimethyl-pyrazine, and 2-ethenyl-6methyl-pyrazine was increased in cocoa beans with enzymatic hydrolysis and fermentation. Tetramethyl pyrazine was the most abundant pyrazine described as having roasted, cocoa, and chocolate notes [51]. Well-fermented cocoa beans were reported to contain higher pyrazine compounds, particularly tetramethyl pyrazine and trimethyl pyrazine [5]. These pyrazines are formed by the amino acid leucine, glycine, alanine, valine, and isoleucine in the Maillard reaction. Additionally, the reaction between the carbonyl group and the amino group of glycine produced primarily 2,5-dimethyl-pyrazine, 2-ethyl-5-methyl-pyrazine, and trimethyl-pyrazine [52].

Enzymatic hydrolysis increased benzaldehyde, benzene acetaldehyde, and 2-phenyl-2-butenal in cocoa beans compared to unfermented cocoa beans, as shown in Table 4. Benzaldehyde and 2-phenyl-2-butenal increased after the fermentation and roasting [53]. High levels of aldehyde compounds are desirable for cocoa quality, and the main aldehyde in roasted fermented cocoa beans was benzaldehyde [54], with sweet, almond, and cherry notes [55]. Flavor compounds such as aldehydes are derived from hydrophobic amino acids, including alanine, valine, leucine, isoleucine, and phenylalanine [56].

The percentage of the total esters with enzymatic hydrolysis was higher than in unfermented cocoa but lower than in fermented cocoa beans. Enzymatic hydrolysis and fermentation increased isoamyl acetate and 1-methoxy-2-propyl acetate in cocoa beans (Table 4). Furthermore, esters are another key flavor of cocoa with fruity, floral, and sweet notes [57], and their total concentration increases during fermentation, drying, and roasting [58]. In fermented cocoa beans, after roasting, 2-phenyl ethyl acetate with fruity, sweet honey, floral and flowery notes was the most abundant in the ester group [54]. Isoamyl acetate is volatile, with fruity notes derived from the free amino acid leucine [59].

Enzymatic hydrolysis and fermentation increased acetic acid, isopentanoic acid, hexanoic acid, and propanoic acid in cocoa beans. Acetic acid is generally described as having sour, astringent, and vinegar notes [60]. The total acid compounds increased during the fermentation and drying process [54]. Microbial activity breaks down sugars to produce acetic acid, which diffuses into the beans [59]. On the other hand, the increase in the enzymatic hydrolysis treatment is due to the acetic acid used for incubation diffused into the beans and changed the acidity of the beans.

**Table 4 foods-12-00820-t004:** Volatile compounds of cocoa beans hydrolyzed by bromelain.

Compound	Detected in Sample (% Area)				Odor Description	Reference
	Non-Fermented	Fermented	Bromelain Treatments			
			(3.5 U/mL, 8 h)	(10.5 U/mL, 6 h)		
**Pyrazines**						
Tetramethyl-pyrazine	0.10	9.58	4.91	5.08	Chocolate, cocoa, coffee	[59]
2,5-Dimethyl-pyrazine	1.17	1.20	0.58	0.66	Cocoa, roasted nuts	[61]
Trimethyl-pyrazine	0.22	2.94	1.45	1.29	Earthy, cocoa, roasted	[53]
3,5-Diethyl-2-methyl-pyrazine	0.07	0.21	0.08	0.05		
2,6-Dimethyl- pyrazine	0.21	0.75	0.23	0.36	Nutty, coffee, green	[53]
2-Ethyl-6-methyl-pyrazine	0.06	0.27	0.10	0.09		
2-Ethyl-5-methyl-pyrazine	1.01	0.68	0.58	0.71	Roasted, green, cocoa	[6]
2-Ethenyl-6-methyl-pyrazine	0.04	0.18	0.15	0.17		
Methyl-pyrazine	0.49	0.61	0.49	0.54	Nutty, cocoa, chocolate, roasted	[61]
2,6-Diethyl-pyrazine	0.34	0.60	0.32	0.38		
2,3-Dimethyl-pyrazine	0.09	0.55	0.33	0.22	Caramel, cocoa, sweet	[53]
2-Acetyl-3-methylpyrazine	0.23	0.24	0.15	0.29		
Ethyl-pyrazine	0.25	0.22	0.25	0.12	Peanut-butter, musty, nutty	[53]
Total	4.28	18.03	9.62	9.96		
**Aldehydes**						
Benzaldehyde	0.17	2.19	1.90	0.52	Sweet, almond, cherry	[55]
Acetaldehyde	4.77	2.06	3.51	3.20	Fresh and fruity	[62]
2-Methyl-butanal	4.23	1.76	3.73	2.99		
Benzeneacetaldehyde	0.42	0.78	1.48	2.85	Almond, fruity, nutty	[63]
2-Isopropyl-5-methyl-2-hexenal	0.11	0.26	0.17	0.10		
2-Phenyl-2-butenal	0.02	0.56	0.13	0.40	Flowery, cocoa, roasted,	[53]
5-Methyl-2-phenyl-2-hexenal	-	0.34	-	0.02		
Hexanal	0.13	0.03	0.08	0.07	Green, fruity, and woody	[55]
Octanal	0.12	-	0.10	0.09		
Total	9.97	7.98	11.10	10.24		
**Esters**						
Isoamyl acetate	0.40	2.28	2.89	2.26	Banana, pear, fruity	[54]
Sec-pentyl acetate	1.03	0.56	0.75	0.74		
1-Methoxy-2-propyl acetate	0.03	1.73	1.17	1.72	Floral honey, rosy, chocolate, and cocoa	[64]
5-Methylfurfuryl acetate	0.03	-	0.09	0.07		
Pentyl benzoate	0.12	-	0.04	0.28		
2-Phenylethyl acetate	-	1.01	-	-	Fruity, sweet honey, floral, flowery	[54]
Ethyl phenylacetate	0.26	0.47	0.16	0.34	Sweet, honey	[61]
Methylglycol acetate	0.53	0.54	0.24	0.70		
Ethyl caprylate	0.06	0.36	0.05	0.06		
Methyl formate	1.83	0.37	0.79	0.38	Fruity	[65]
1,2-Propanediol diformate	0.03	0.23	0.10	0.06		
Allyl 2-ethyl butanoate	-	0.16	-	-		
Butyl acetate	-	0.12	-	-	Fruity, apple, banana	[59]
Ethyl butanoate	0.94	0.08	0.68	0.27		
Isobutyl acetate	0.10	0.07	0.19	0.20	Fruity	[53]
Hexyl acetate	0.13	0.02	0.04	0.05		
Butyl propanoate	0.33	-	-	-		
Methyl hexanoate	0.28	-	-	-		
Ethyl 3-hydroxybutyrate	0.10	-	0.07	0.16		
Total	6.20	8.00	7.26	7.29		
**Alcohols**						
Phenylethyl alcohol	1.30	3.31	1.80	2.27	Floral, sweet, and bready	[60]
2-Nonanol	0.05	0.60	0.18	0.48	Fat, green	[66]
1-Butanol	0.94	0.14	0.64	0.21		
2-Octanol	-	0.55	-	-	Fresh, spicy green, earthy	[55]
2-Methyl-3-pentanol	0.14	0.16	0.14	0.09		
Isoamyl alcohol	-	0.55	-	-	Balsamic fruit	[55]
Benzyl alcohol	0.33	0.44	0.18	0.40	Sweet, fruity	[53]
3-Methyl-2-heptanol	0.39	0.31	0.14	0.46		
Ethanol	4.82	0.28	0.25	0.20	Alcoholic, Pungent	[62]
1,3-Butanediol	-	0.07	-	-		
Isobutanol	-	0.03	-	-		
2-Heptanol	1.45	0.27	1.01	0.57	Citrusy, sweet, fruity, lemon grass	[53,59]
2-Dodecanol	0.17	-	-	-		
2-Nonadecanol	0.19	-	0.07	0.34		
6-Methyl-2-heptanol	0.26	0.27	0.10	0.00	Citrus, fruity, lemon grass	[59]
2-Pentanol	17.14	0.26	0.88	0.38	Mild and green	[60]
2-Ethoxy-1-propanol	0.03	0.26	0.15	0.31		
1-Pentanol	-	0.21	-	-		
2,3-Butanediol	-	0.20	-	-	Sweet, flowery	[53]
Isobutylmethylcarbinol	0.21	-	-	-		
2-Methyl-1,3-butanediol	0.05	-	-	-		
Acetylcarbinol	0.44	0.19	0.48	0.51		
Diisobutylcarbinol	0.06	0.12	0.15	0.16		
Vinyldimethylcarbinol	0.60	0.07	-	-		
Total	28.57	8.29	6.17	6.38		
**Acids**						
Acetic acid	10.98	28.78	33.08	32.81	Strong, pungent	[61]
Isopentanoic acid	1.56	6.69	7.45	3.29	Sweat, acid and rancid	[60]
Benzeneacetic acid	-	0.10	-	-		
Isobutyric acid	-	3.48	-	-	Rancid, buttery, cheesy	[61]
3-Hydroxyisovaleric acid	-	0,10	-	-		
Hexanoic acid	0.46	0.48	0.26	0.43	Pungent, sickening, rancid, sour	[61]
Isohexanoic acid	0.18	0.47	0.24	0.35		
Propanoic acid	0.15	0.30	0.22	0.21	Pungent and rancid	[51]
Total	13.33	40.40	41.25	37.09		
**Ketones**						
Acetoin	0.07	1.47	1.59	1.47	Buttery and creamy	[53]
3-Hexanone	0.28	0.06	0.21	0.12		
2-Methyloxolan-3-one	0.20	0.09	0.13	0.06		
Acetophenone	1.11	0.58	0.80	1.00	Floral and sweet	[53]
2-Butanone	0.83	0.56	0.55	0.47		
2-Octen-4-one	-	0.06		-		
2-Nonanone	0.08	0.40	0.10	0.23	Flowery, fatty	[53]
Methylcyclohexenone	0.10	0.38	0.23	0.20		
2-Heptanone	0.61	0.37	0.70	0.60	Fruity, green, flowery	[53]
2,3-Octanedione	0.43	-	0.07	0.09		
2-Pentanone	0.78	0.24	0.24	0.30	Fruity	[55]
2-Acetoxy-3-butanone	0.02	0.16	0.17	0.07		
3-Ethylidene-heptan-2,6-dione	0.04	0.07	0.02	0.01		
N-Methyloxazolidone	-	0.03	-	-		
6-Methyl-5-hepten-2-one	0.09	-	-	-		
1-Methylpyrrolidinone	0.04	-	-	-		
Dimethyl sulfone	0.13	-	-	-		
Total	4.81	4.48	4.81	4.62		

However, the flavor obtained in this study, such as pyrazines and esters, which are the primary flavors, was lower than that of fermented cocoa beans. This might be due to the lower reducing sugars and higher polyphenols in cocoa beans with enzymatic hydrolysis. The decrease in flavor compounds can be caused by the binding of polyphenols with flavor precursors or volatile compounds formed during roasting. Polyphenols can inhibit the Maillard reaction by trapping intermediate products, such as α-dicarbonyl, during the roasting process, resulting in less formation of flavor compounds [34,47].

### 3.6. Principal Component Analysis

PCA was performed to describe the correlation between parameters such as the degree of hydrolysis, free amino acids, and volatile compounds with different treatments on unfermented cocoa beans. The principal component (PC) explains about 83.38% of the total variability of all data, with the first main component (PC) contributing 63.12% and the second PC contributing 20.26% as shown in Figure 5. The relationship between parameters and the principal component factors is shown in Table 5, which shows the magnitude of load that reflected the importance of each variable in the principal components. The degree of hydrolysis (%) and the valine, leucine, isoleucine, tyrosine, lysine, threonine, and volatile esters have a high load on PC 1, while the glycine and aldehydes have a high load on PC 2, and glutamic acid has a high load on PC 3. Moreover, polyphenols, amino acid glycine, aldehydes, alcohols, and ketones negatively affect PC 1.

From the biplot (Figure 5) based on the variable contribution to PC 1 and 2, we can see that there is a correlation between the treatment of unfermented cocoa beans and volatile alcohol. Volatile alcohols with off-flavors, such as ethanol and 2-methyl-1-propanol, and ketones are high in unfermented cocoa beans and decrease during fermentation [59]. On the other hand, the degree of hydrolysis, reducing sugars, and free amino acids as flavor precursors and desired volatile compounds such as pyrazines and esters were positively correlated with hydrolysis treatment using a bromelain concentration of 10.5 U/mL for 6 h and fermented cocoa beans. This indicates that fermentation and hydrolysis treatment with bromelain on unfermented cocoa beans has an impact on increasing flavor precursors and the desired flavor of unfermented cocoa beans. In previous studies, an increase in DH was observed in hydrolysis with bromelain [14,16,17]. There was an increase in free amino acids in fermented cocoa beans due to enzymatic hydrolysis by the endogenous carboxypeptidase and aspartate endoproteinase [30], whereas, in the bromelain treatment, bromelain could break down proteins to produce high free amino acids [14,17,38]. Furthermore, the increase of flavor precursors in fermented cocoa beans and hydrolysis treatment with bromelain increased the desired flavor [48].

## 4. Conclusions

The processing methods of cocoa beans affect the development of flavor precursors and cocoa-bean flavors. Free amino acids and reducing sugars are flavor precursors that produce volatile compounds such as pyrazines, aldehydes, and esters during roasting by the Maillard reaction.

Based on the results, enzymatic hydrolysis treatment resulted in higher free amino acids than unfermented cocoa beans, especially hydrophobic amino acids, such as phenylalanine, valine, leucine, alanine, and tyrosine, mainly due to the protein breakdown by bromelain and the role of endogenous enzymes. The percentage of reducing sugars in the enzymatic hydrolysis treatment was also higher (7.34–8.16%) than that of unfermented cocoa beans (4.60%), possibly due to the activation of endogenous enzymes during hydrolysis. On the other hand, the concentration of polyphenols with a negative impact on cocoa flavor decreased in the enzymatic hydrolysis treatment (20.80–19.75 mg GAE/g). These results indicate that the enzymatic hydrolysis treatment produces better flavor precursors for flavor development than unfermented cocoa beans as a control.

With the increase in flavor precursors, there was an increase in the desired volatile compounds in cocoa beans by enzymatic hydrolysis, resulting in more abundant cocoa flavors. However, the primary flavors obtained were lower than fermented cocoa beans, such as pyrazines and esters, and this is probably due to the lower reducing sugars and higher polyphenols. Furthermore, additional studies are needed to increase reducing sugars as flavor precursors and reduce polyphenols that affect cocoa flavors.

The PCA results showed that enzymatic hydrolysis treatment with the bromelain concentration of 10.5 U/mL for 6 h had a positive correlation with the degree of hydrolysis, free amino acids, and desired cocoa flavors. Therefore, hydrolysis with a bromelain concentration of 10.5 U/mL for 6 h seems to be the optimal condition for precursor and flavor development in unfermented cocoa beans.

## Figures and Tables

**Figure 1 foods-12-00820-f001:**
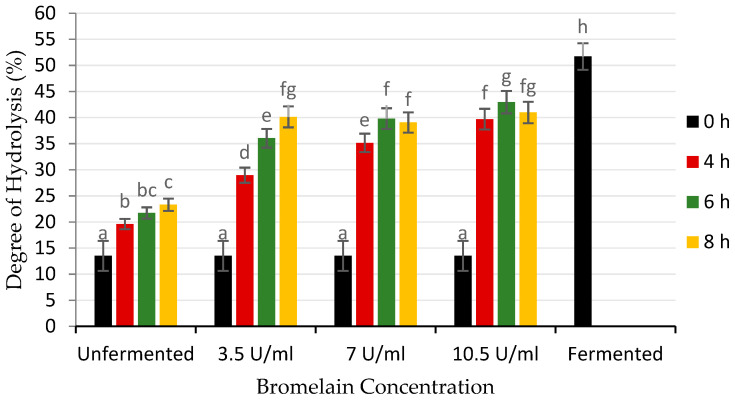
Degree of hydrolysis (%) of unfermented cocoa beans hydrolyzed by bromelain. Unfermented and fermented cocoa beans were used as a control. ^a–g^ Non-capital letters represent the statistical differences in the interaction of enzyme concentration and hydrolysis time (*p* < 0.05).

**Figure 2 foods-12-00820-f002:**
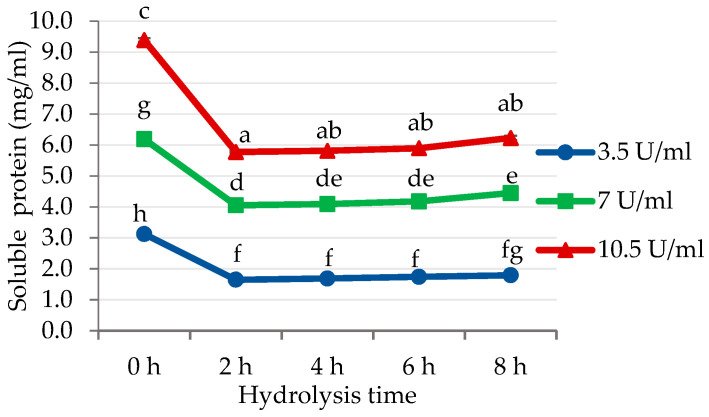
Soluble protein in the buffer–enzyme solution during hydrolysis with bromelain. ^a–h^ non-capital letters represent the statistical differences (*p* < 0.05).

**Figure 3 foods-12-00820-f003:**
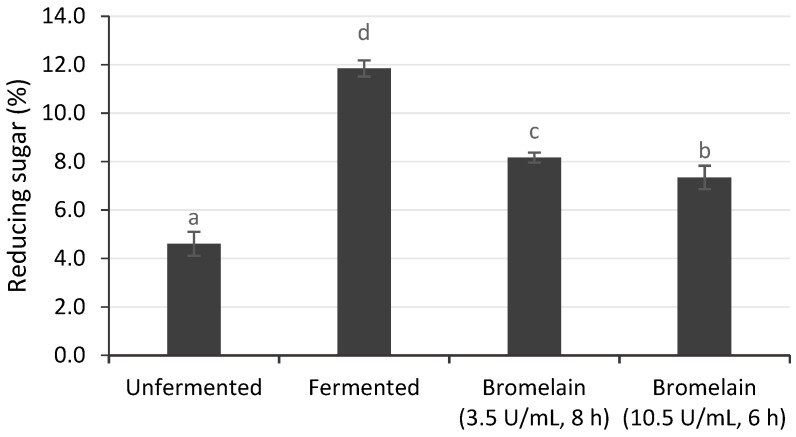
Reducing sugar of unfermented cocoa beans hydrolyzed by bromelain. Unfermented and fermented cocoa beans were used as controls. ^a–d^ Indicates the significant difference (*p* < 0.05).

**Figure 4 foods-12-00820-f004:**
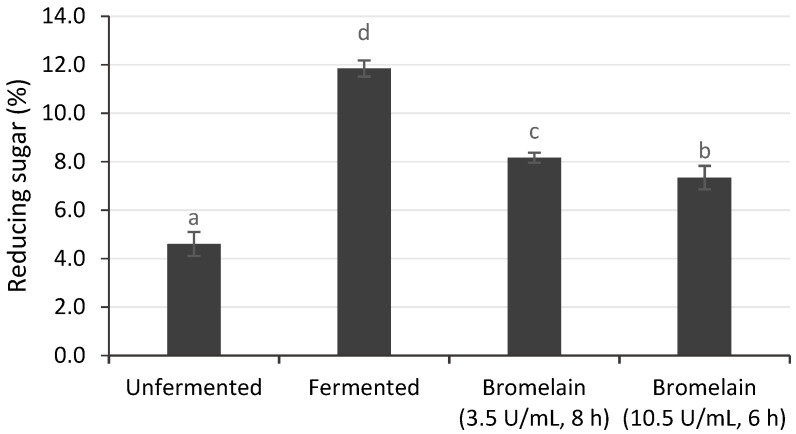
Total polyphenols of unfermented cocoa beans hydrolyzed by bromelain. Unfermented and fermented cocoa beans were used as controls. ^a–d^ Indicate the significant difference (*p* < 0.05).

**Figure 5 foods-12-00820-f005:**
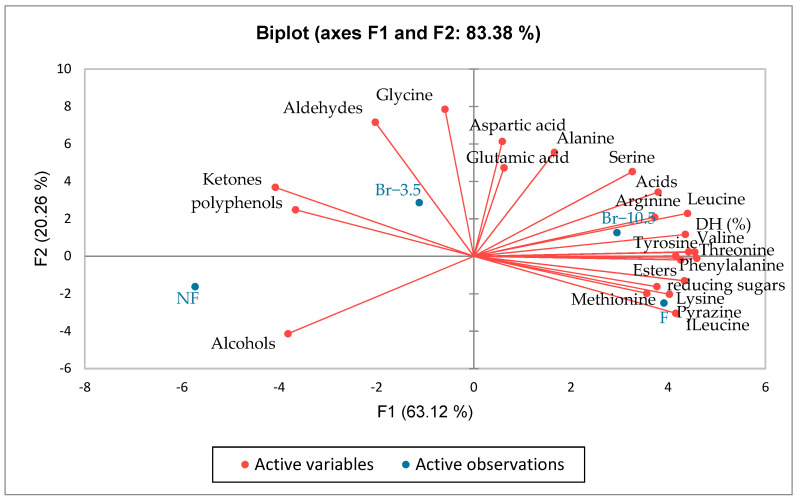
Principal component analysis (PCA) biplot for correlations between parameters of the degree of hydrolysis, free amino acids, reducing sugars, polyphenols, free amino acids, and volatile compounds with different treatments of unfermented cocoa beans: NF = unfermented cocoa beans; Br−3.5 = bromelain concentrations of 3.5 U/mL for 8 h; Br−10.5 = bromelain concentrations of 10.5 U/mL for 6 h; F = fermented cocoa beans.

**Table 1 foods-12-00820-t001:** Bromelain enzyme activity at various temperatures.

pH	Temperature (°C)	Enzyme Activity (U/mL)	Specific Activity (U/mg Protein Enzyme)
7	40	0.065 ± 0.005 ^a^	0.34 ± 0.03
	45	0.109 ± 0.008 ^c^	0.57 ± 0.04
	50	0.205 ± 0.010 ^d^	1.18 ± 0.06
	55	0.190 ± 0.004 ^d^	1.00 ± 0.02
	60	0.094 ± 0.003 ^b^	0.50 ± 0.02

The values represented mean ± SD. Different letters indicate a significant difference (*p* < 0.05).

**Table 2 foods-12-00820-t002:** Bromelain enzyme activity at various pH values.

Temperature	pH	Enzyme Activity (U/mL)	Specific Activity (U/mg Protein Enzyme)
50 °C	5	0.073 ± 0.007 ^a^	0.35 ± 0.03
	5.5	0.160 ± 0.003 ^b^	0.84 ± 0.02
	6	0.210 ± 0.021 ^c^	1.11 ± 0.04
	6.5	0.206 ± 0.006 ^c^	1.09 ± 0.03
	7	0.021 ^c^	1.08 ± 0.02

The values represented mean ± SD. Different letters indicate a significant difference (*p* < 0.05).

**Table 3 foods-12-00820-t003:** Free amino acid of cocoa beans hydrolyzed by bromelain.

Free Amino Acids	Sample (µg/g)			
	Unfermented	Fermented	Bromelain Treatments	
			(3.5 U/mL, 8 h)	(10.5 U/mL, 6 h)
leucine	19.29 ± 0.04	25.38 ± 0.03	23.96 ± 0.03	25.78 ± 0.04
L-iso-leucine	10.19 ± 0.04	12.37 ± 0.02	10.21 ± 0.02	11.93 ± 0.01
phenylalanine	16.22 ± 0.01	22.39 ± 0.01	19.08 ± 0.03	21.76 ± 0.03
valine	12.65 ± 0.03	15.52 ± 0.02	13.95 ± 0.01	15.63 ± 0.02
alanine	24.01 ± 0.01	23.82 ± 0.02	25.44 ± 0.05	27.73 ± 0.01
tyrosine	22.49 ± 0.02	27.06 ± 0.06	23.69 ± 0.02	28.88 ± 0.05
glycine	24.52 ± 0.07	22.45 ± 0.02	27.83 ± 0.06	27.10 ± 0.03
arginine	23.85 ± 0.03	27.53 ± 0.04	25.89 ± 0.01	31.45 ± 0.02
lysine	21.98 ± 0.02	24.21 ± 0.03	22.38 ± 0.02	24.33 ± 0.03
serine	22.57 ± 0.06	24.65 ± 0.03	25.28 ± 0.05	28.06 ± 0.02
threonine	14.28 ± 0.02	16.31 ± 0.02	15.08 ± 0.03	16.69 ± 0.04
methionine	9.83 ± 0.03	11.39 ± 0.04	9.33 ± 0.02	12.15 ± 0.03
aspartic acid	38.54 ± 0.05	41.06 ± 0.07	40.63 ± 0.05	47.20 ± 0.05
glutamic acid	50.49 ± 0.06	57.05 ± 0.05	52.16 ± 0.03	66.37 ± 0.04

The values represented mean ± SD.

**Table 5 foods-12-00820-t005:** Correlations between variables and factors.

Parameters	F1	F2	F3
DH (%)	0.949	0.144	0.282
Reducing sugars (%)	0.821	−0.201	0.534
Polyphenols (mg/g)	−0.799	0.307	−0.517
Leucine	0.958	0.283	0.043
ILeucine	0.905	−0.376	−0.198
Phenylalanine	1.000	−0.013	−0.008
Valine	0.991	0.028	−0.132
Alanine	0.361	0.685	−0.632
Tyrosine	0.906	0.002	−0.423
Glycine	−0.129	0.970	−0.206
Arginine	0.811	0.257	−0.525
Lysine	0.944	−0.162	−0.287
Serine	0.711	0.559	−0.427
Threonine	0.964	0.030	−0.265
Methionine	0.776	−0.245	−0.582
Aspartic acid	0.128	0.757	0.641
Glutamic acid	0.135	0.584	0.801
Pyrazine	0.877	−0.251	0.409
Aldehydes	−0.442	0.885	−0.149
Esters	0.928	−0.025	0.371
Alcohols	−0.833	−0.511	−0.212
Acids	0.827	0.423	0.370
Ketones	−0.891	0.455	−0.010

## Data Availability

Data is contained within the article.

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
