# Peer review of "Flavor Precursors and Volatile Compounds Improvement of Unfermented Cocoa Beans by Hydrolysis Using Bromelain"

_foods, 2023, doi:10.3390/foods12040820_

Round 1
Reviewer 1 Report
The aroma of cocoa is largely determined by the genotype of the variety, the time of harvest, the pre-fermentation treatments, fermentation, drying and roasting. Fresh cocoa beans have a vinegary smell and taste. Changes within the cells of the petals release key enzymes that trigger reactions between the precursors present in the kernel before fermentation. In order to obtain the right organoleptic quality and avoid flavour defects, particular care must be taken during fermentation of the beans and the treatments that precede it are also of great importance.
The precursors responsible for the flavour of chocolate include free amino acids, reducing sugars, polyphenols, volatile compounds formed during fermentation and undergoing the necessary changes during drying and roasting. Fermentation produces volatile acidity of about 2 % of dry matter. These include acetic acid, propionic acid, butyric acid, isobutyric acid and isovaleric acid. In terms of proportion, acetic acid accounts for 90% of these acids and plays a very important role in catalysing the enzymatic processes that produce the desired organoleptic characteristics. Cocoa beans that have not been fermented will not be aromatic and will be very bitter and astringent. Fermentation contributes to the breakdown of the flesh covering the kernel, making drying easier. This process triggers the biochemical changes that form the precursor compounds necessary for colour, aroma and flavour. In some parts of the world, many small-scale producers dry cocoa beans directly, without fermentation, which results in fewer flavour precursors and less cocoa flavour. Therefore, this research is of particular interest, which enhances flavour compounds by hydrolysis of the unfermented cocoa bean flavour precursors, free amino acids and volatile compounds, using bromelain.
The title summarises the research topic adequately and succinctly.
The abstract presents the research topic in a precise way, but it would also be useful to indicate the reasons why some small farmers dry their cocoa beans directly without fermentation. The abstarct presents the objective well, describing the material and method, but the results are mixed.
For the keywords, it is advisable to choose ones that are relevant but not included in the title.
The introduction section briefly summarises the biochemical changes that occur during fermentation. It is possible that not only in Indonesia, but also in other small producers around the world, cocoa beans are dried directly without fermentation. It would be worthwhile to substantiate this with figures, thus increasing the relevance of the research. Overall, it would be worth summarising the most important indicators of world cocoa production in a separate paragraph (volume value/country, thousand tonnes/country, exports, imports, etc.).
The introduction section should be extended with additional international publications that have investigated the enhancement of free amino acids by enzymatic hydrolysis with bromelain.
The manuscript should be supplemented with justification of the measurement parameters for enzymatic hydrolysis and volatile components (2.4 Enzymatic Hydrolysis of Unfermented Cocoa Beans Using Bromelain and 2.10 Volatile Compound Analysis). Conditional tests for the use of ANOVA should be performed and the results reported in the manuscript.
Tables are provided to illustrate the results (Table 1. Bromelain Enzyme Activity at Various Temperatures, Table 2. Bromelain Enzyme Activity at Various pH).
It would be worthwhile to remove the abbreviation 'DH', most appropriately at the first mention (L164).
This result should be supported by several other international results: the highest DH was 42.95% at a bromelain concentration of 10.5 U/mL for 6 hours of hydrolysis, but was not significantly different from the value at a concentration of 3.5 U/mL for 8 hours.
Figures 1 and 2 illustrate the results. In the case of Figure 2 (Soluble Protein in the Buffer-Enzyme Solution during Hydrolysis with Bromelain), it would be useful to indicate significant differences (preferably with ANOVA and post-hoc test using lettering).
The standard deviation values are missing from this table (Table 3. Free amino acid of cocoa beans hydrolyzed by bromelain).
The results for reducing sugars and polynol content are correct and well supported.
The presentation of the volatile components is detailed and correctly grouped (Pyrazines, Aldehydes, Esters, Alcohols, Acids, Ketones) and the volatile equivalents of each component are included. One of the most important of the results is Table 6 (Table 4. Volatile compounds of cocoa beans hydrolysed by bromelain).
The conclusion is too short, it would be better not to repeat the results but to draw conclusions from them. The conclusion section could be extended, for example, with a limitation or further research questions and further studies.
Author Response
Response to Reviewer 1
[General comment] The aroma of cocoa is largely determined by the genotype of the variety, the time of harvest, the pre-fermentation treatments, fermentation, drying and roasting. Fresh cocoa beans have a vinegary smell and taste. Changes within the cells of the petals release key enzymes that trigger reactions between the precursors present in the kernel before fermentation. In order to obtain theright organoleptic quality and avoid flavour defects, particular care must be taken during fermentation of the beans and the treatments that precede it are also of great importance
The precursors responsible for the flavour of chocolate include free amino acids, reducing sugars, polyphenols, volatile compounds formed during fermentation and undergoing the necessary changes during drying and roasting. Fermentation produces volatile acidity of about 2 % of dry matter. These include acetic acid, propionic acid, butyric acid, isobutyric acid and isovaleric acid. In terms of proportion, acetic acid accounts for 90% of these acids and plays a very important role in catalysing the enzymatic processes that produce the desired organoleptic characteristics. Cocoa beans that have not been fermented will not be aromatic and will be very bitter and astringent. Fermentation contributes to the breakdown of the flesh covering the kernel, making drying easier. This process triggers the biochemical changes that form the precursor compounds necessary for colour, aroma and flavour. In some parts of the world, many small-scale producers dry cocoa beans directly, without fermentation, which results in fewer flavour precursors and less cocoa flavour. Therefore, this research is of particular interest, which enhances flavour compounds by hydrolysis of the unfermented cocoa bean flavour precursors, free amino acids and volatile compounds, using bromelain.
Response: thank you, we realized that there were several sentences that had not been added in the introduction, so we have added a sentence in lines 30-32 as follow:
“The development of cocoa flavor is largely determined by the genetic profile of cocoa beans, the growing environment and the processing methods used such as fermentation and drying [2]”
And in line 32-34 as follow:
“Fermentation produces volatile acidity of about 2 % of dry matter. Acetic acid accounts for 90% of the total acids and plays an important role in catalyzing the enzymatic processes to develop flavor precursors [3]”
[Comment-1] The title summarises the research topic adequately and succinctly.
Response: Thank you for the comment, we have tried to make a title that can summarises the research topic adequately and succinctly.
[Comment-2] The abstract presents the research topic in a precise way, but it would also be useful to indicate the reasons why some small farmers dry their cocoa beans directly without fermentation.
Response: we have added a sentence in our abstract to indicate why some small farmers dry their cocoa beans directly without fermentation as follow:
“Cocoa fermentation is an essential process that produces flavor precursors. However, many small farmers in Indonesia directly dry their cocoa without fermentation due to low yield and long fermentation time, resulting in fewer flavor precursors and cocoa flavor”
[Comment-3] For the keywords, it is advisable to choose ones that are relevant but not included in the title.
Response: We thank reviewers for this advice. we have revised our keyword as follows: Enzymatic hydrolysis; Unfermented Cocoa Beans, Free amino acids, Cocoa Flavor.
[Comment-4] The introduction section briefly summarises the biochemical changes that occur during fermentation. It is possible that not only in Indonesia, but also in other small producers around the world, cocoa beans are dried directly without fermentation. It would be worthwhile to substantiate this with figures, thus increasing the relevance of the research.
Response: We have added a few sentences in the Introduction (Lines 50-53) as follows:
Many small Indonesian farmers dry fresh cocoa beans without fermentation due to low yield, long fermentation time, and a price difference that is not significant [9,10] “In addition, many small producers in other countries also dried their cocoa beans without the fermentation process. About 30% of small farmers in Ecuador sell cocoa beans without fermentation and drying processes [11]”
[Comment-5] Overall, it would be worth summarising the most important indicators of world cocoa production in a separate paragraph (volume value / country, thousand tonnes/country, exports, imports, etc.).
Response: We have added a few sentences in the Introduction (Lines 42-48) as follows:
“African countries such as Côte d'Ivoire, Ghana, and Cameroon are the major producers of cocoa beans supplying 74.5% of the global production, with a total cocoa production of 3.6 million tons in 2021. Meanwhile, Asian countries supply 5.5% of global production with Indonesia producing 200 thousand tons of cocoa in 2021 [7].
After Ivory Coast and Ghana, Indonesia is the third largest exporter of cocoa beans. About 88.48% of cocoa beans in Indonesia are managed by small farmers, and almost 80% of cocoa production is exported to the international market [8]”
[Comment-6] The introduction section should be extended with additional international publications that have investigated the enhancement of free amino acids by enzymatic hydrolysis with bromelain.
Response: we have added additional international publications that have investigated the enhancement of free amino acids by enzymatic hydrolysis with bromelain (Lines 67-70) as follows:
“A previous study found that enzymatic hydrolysis using bromelain in mung bean, brown rice and Seaweed (G. fisheri) by-products resulted in higher hydrophobic amino acids, which are significant in the formation of flavor compounds [14, 16, 17]”
[Comment-7] The manuscript should be supplemented with justifications of the measurement parameters for enzymatic hydrolysis and volatile components (2.4 Enzymatic Hydrolysis of Unfermented Cocoa Beans Using Bromelain and 2.10 Volatile Compound Analysis). Conditional tests for the use of ANOVA should be performed and the results reported in the manuscript.
Response: Justification of the measurement parameters for volatile components at 2.10 Volatile Compound Analysis as follows: The volatile aroma compounds were analyzed based on a modified method described by Caprioli et al., (2016).
and Enzymatic Hydrolysis of Unfermented Cocoa Beans Using Bromelain (2.4) is a research method designed by the authors.
ANOVA has been used in this study such as in the analysis of the degree of hydrolysis, reducing sugars, total polyphenols whose results are reported in the manuscript.
[Comment-8] Tables are provided to illustrate the results (Table 1. Bromelain Enzyme Activity at Various Temperatures, Table 2. Bromelain Enzyme Activity at Various pH).
Response:
Table 1. Bromelain Enzyme Activity at Various Temperatures, Table 2. Bromelain Enzyme Activity at Various pH are provided to illustrate the results described in lines 154-160 as follows:
“The highest bromelain activity was found at 50°C, as shown in Table 1, bromelain activity increased with increasing temperature up to 50°C and then decreased. And According to Table 2, the highest enzyme activity was obtained at pH 6, as the activity increased with increasing pH up to 6 and then decreased”
[Comment-9] It would be worthwhile to remove the abbreviation 'DH', most appropriately at the first mention (L164).
Response:
we have revised "DH" to "degree of hydrolysis" in line 178-179:
“Figure 1 shows the degree of hydrolysis of unfermented, fermented, and unfermented cocoa beans hydrolyzed by bromelain”
[Comment-10] This result should be supported by several other international results: the highest DH was 42.95% at a bromelain concentration of 10.5 U/mL for 6 hours of hydrolysis, but was not significantly different from the value at a concentration of 3.5 U/mL for 8 hours.
Response: we have added some other international results (lines 212-215) to support our results as follows:
“In previous studies, hydrolysis of mung beans with bromelain at a concentration of 20% (w/w) for 6 hours produced the highest DH of 50.4% but was not significantly different from 15% (w/w) at 12, 18, and 24 hours [16]. In the hydrolysis of seaweed protein by-products using bromelain resulted in an increase in DH and reaching a plateau after 6 hours with a DH of 62.91%. However, DH values ​​of 15% (w/w) and 20% (w/w) were not significantly different [14]”
[Comment-11] Figures 1 and 2 illustrate the results. In the case of Figure 2 (Soluble Protein in the Buffer-Enzyme Solution during Hydrolysis with Bromelain), it would be useful to indicate significant differences (preferably with ANOVA and post-hoc test using lettering).
Response: we have added ANOVA and post hoc test using lettering in Figure 2 (Soluble Protein in the Buffer-Enzyme Solution during Hydrolysis with Bromelain) as follows:
[Comment-12] The standard deviation values are missing from this table (Table 3. Free amino acid of cocoa beans hydrolyzed by bromelain).
Response: we have added the standard deviation values in Table 3. free amino acid of cocoa beans hydrolyzed by bromelain
[Comment-13] The conclusion is too short, it would be better not to repeat the results but to draw conclusions from them. The conclusion section could be extended, for example, with a limitation or further research questions and further studies.
Response: We thank reviewers for this suggestion. we have revised the conclusions as follows:
Processing methods of cocoa beans affect the development of flavor precursors and cocoa bean flavors. Free amino acids and reducing sugars are flavor precursors that produce volatile compounds such as pyrazines, aldehydes, and esters during roasting by the Maillard reaction.
Based on the results, enzymatic hydrolysis treatment resulted in higher free amino acids than unfermented cocoa beans, especially hydrophobic amino acids, such as phenylalanine, valine, leucine, alanine, and tyrosine mainly due to protein breakdown by bromelain and the role of endogenous enzymes. Reducing sugars in the enzymatic hydrolysis treatment was also higher (7.34-8.16%) than unfermented cocoa beans (4.60%), possibly due to the activation of endogenous enzymes during hydrolysis. On the other hand, the concentration of polyphenols with a negative impact on cocoa flavor decreased in the enzymatic hydrolysis treatment (20.80- 19.75 mg GAE/g). These results indicate that the enzymatic hydrolysis treatment produces better flavor precursors for flavor development than unfermented cocoa beans as a control.
By increasing flavor precursors, there was an increase in the desired volatile compounds in cocoa beans by enzymatic hydrolysis, resulting in more abundant cocoa flavors. However, the primary flavors obtained were lower than fermented cocoa beans, such as pyrazines and esters, which is probably due to the lower reducing sugars and higher polyphenols. Furthermore, additional studies are needed to increase reducing sugars as precursor flavors and reduce polyphenols that affect cocoa flavors.
PCA results showed that enzymatic hydrolysis treatment with the bromelain con-centration of 10.5 U/ml for 6 hours had a positive correlation with the degree of hydrolysis, free amino acids, and desired cocoa flavors. Therefore, hydrolysis with the bromelain con-centration of 10.5 U/ml for 6 hours seems to be the optimal condition for precursor and flavor development in unfermented cocoa b

Reviewer 2 Report
The paper “Flavor Precursors and Volatile Compounds Improvement of Unfermented Cocoa Beans by Hydrolysis Using Bromelain” contributes to the growth of literature for nutritionists and food producers offering Cocoa Beans-based products.
The following items should be revised:
Line 33-34
“Additionally, the polyphenol decrease due to oxidation by polyphenol oxidase [1].” what polyphenols?
Statistical Analysis
I suggest extending the statistical analysis by Principal component analysis (PCA) and Correlation coefficients between variable hydrolysis parameters, e.g. temperature, time, Degree of Hydrolysis and free amino acid
also a correlation between total polyphenol, free amino acids, variable hydrolysis parameters and Volatile compounds.
Table 4
content of, e.g. acids 13,3 - good value? should it be 13.3?
Similarly, the conclusions: "Based on the results, enzymatic hydrolysis of unfermented cocoa beans increased the degree of hydrolysis to 76,98 – 83,07% of fermented cocoa beans and resulted in higher".
The authors did not summarize their results. The authors should add the summary conclusion - the positive or negative effects of the research.
What are the limitations of this research? - high content of reducing sugars and high temperature in the products?
The Maillard reaction generates the desirable aroma, but may also be associated with the formation of potentially toxic compounds such as acrylamide.
The research on Volatile compounds is not supported by sensory analysis.
Author Response
Response to Reviewer 2
[General comment] The paper “Flavor Precursors and Volatile Compounds Improvement of Unfermented Cocoa Beans by Hydrolysis Using Bromelain” contributes to the growth of literature for nutritionists and food producers offering Cocoa Beans-based products.
Response: Thank you for the comment about this paper
[Comment-1] Line 33-34 “Additionally, the polyphenol decrease due to oxidation by polyphenol oxidase [1].” What polyphenols?
Response: We have added a sentence in introduction (Line 37-38)to explain what polyphenols are in cocoa beans as follows :
“Additionally, the polyphenols decrease due to oxidation by polyphenol oxidase [1]. The main phenolic compounds in cocoa beans are flavan-3-ols (epicatechin and catechin), anthocyanins and flavanols [4]”
[Comment-2] Statistical Analysis I suggest extending the statistical analysis by Principal component analysis (PCA) and Correlation coefficients between variable hydrolysis parameters, e.g. temperature, time, Degree of Hydrolysis and free amino acid also a correlation between total polyphenol, free amino acids, variable hydrolysis parameters and Volatile compounds.
Response: We thank the reviewers for this suggestion. we have added the statistical analysis by Principal component analysis (PCA) and Correlation coefficients between variable hydrolysis parameters to this paper.
[Comment-3] Table 4 content of, e.g. acids 13,3 - good value? should it be 13.3 and Similarly, the conclusions: "Based on the results, enzymatic hydrolysis of unfermented cocoa beans increased the degree of hydrolysis to 76,98 – 83,07% of fermented cocoa beans and resulted in higher".
Response: we have revised 13,3 to 13.3 (table 4) and the conclusion as follows:
Total acid
|
Total |
13.33 |
40.47 |
41.26 |
37.10 |
Based on the results, enzymatic hydrolysis of unfermented cocoa beans increased the degree of hydrolysis to 76.98 – 83.07% of fermented cocoa beans and resulted in higher
[Comment-4] The authors did not summarize their results. The authors should add the summary conclusion - the positive or negative effects of the research.
Response: We thank the reviewer for this suggestion. we have revised the conclusions as follows:
Processing methods of cocoa beans affect the development of flavor precursors and cocoa bean flavors. Free amino acids and reducing sugars are flavor precursors that produce volatile compounds such as pyrazines, aldehydes, and esters during roasting by the Maillard reaction.
Based on the results, enzymatic hydrolysis treatment resulted in higher free amino acids than unfermented cocoa beans, especially hydrophobic amino acids, such as phenylalanine, valine, leucine, alanine, and tyrosine mainly due to protein breakdown by bromelain and the role of endogenous enzymes. Reducing sugars in the enzymatic hydrolysis treatment was also higher (7.34-8.16%) than unfermented cocoa beans (4.60%), possibly due to the activation of endogenous enzymes during hydrolysis. On the other hand, the concentration of polyphenols with a negative impact on cocoa flavor decreased in the enzymatic hydrolysis treatment (20.80- 19.75 mg GAE/g). These results indicate that the enzymatic hydrolysis treatment produces better flavor precursors for flavor development than unfermented cocoa beans as a control.
By increasing flavor precursors, there was an increase in the desired volatile compounds in cocoa beans by enzymatic hydrolysis, resulting in more abundant cocoa flavors. However, the primary flavors obtained were lower than fermented cocoa beans, such as pyrazines and esters, which is probably due to the lower reducing sugars and higher polyphenols. Furthermore, additional studies are needed to increase reducing sugars as precursor flavors and reduce polyphenols that affect cocoa flavors.
PCA results showed that enzymatic hydrolysis treatment with the bromelain con-centration of 10.5 U/ml for 6 hours had a positive correlation with the degree of hydrolysis, free amino acids, and desired cocoa flavors. Therefore, hydrolysis with the bromelain con-centration of 10.5 U/ml for 6 hours seems to be the optimal condition for precursor and flavor development in unfermented cocoa beans.
[Comment-5] What are the limitations of this research? - high content of reducing sugars and high temperature in the products?
Response: The limitations of this study were the flavor obtained such as pyrazines and esters which are the main flavors lower than fermented cocoa beans. This may be due to lower reducing sugars and higher polyphenols in cocoa beans with enzymatic hydrolysis compared to fermented cocoa beans.
Free amino acids and reducing sugars are flavor precursors that produce volatile compounds such as pyrazines, aldehydes, and esters during roasting [5,6]. On the other hand, High concentrations of polyphenols negatively impact cocoa flavor. the concentrations of precursor flavors and flavors formed during roasting decreased with higher concentrations of polyphenols [34,48].
In this study, reducing sugar in cocoa beans with enzymatic hydrolysis was significantly higher ((7.34% and 8.16%) than unfermented cocoa beans (4.60%) but lower than fermented cocoa beans (11.85%) as shown in Figure 3. And the total polyphenol of cocoa beans by enzymatic hydrolysis was lower (20.80 and 19.75 mg GAE/g) than unfermented cocoa beans (25.08 mg GAE/g), but higher than fer-mented cocoa beans (11.70 mg GAE/g) (Figure 4).
[Comment-6] The Maillard reaction generates the desirable aroma, but may also be associated with the formation of potentially toxic compounds such as acrylamide.
Response: The Maillard reaction produces the desired flavor but is also associated with the formation of potentially toxic compounds such as acrylamide. At high temperatures, acrylamide formation was reached in a short time. Increased concentrations of acrylamide were also associated with increased pH and higher asparagine amino acids (Toro et al.,2022).
In this study, the amino acid asparagine was very low so it was not detected. This allows for the formation of less acrylamide in the hydrolysis treatment of cocoa beans with bromelain during roasting.
[Comment-7] The research on Volatile compounds is not supported by sensory analysis.
Response: We thank the reviewer for this advice. In this research we focused on the development of flavor precursors and desired volatile compounds as a flavor of roasted cocoa beans, Therefore, sensory analysi

Reviewer 3 Report
This is a well-written, straightforward, but somewhat premature paper on the use of a peptidase for the pre-treatment of non-fermented cocoa beans with the goal to improve the flavor profile. Similar work, for example using the cheap peptidase mix Flavourzyme from Novo, has been done before. However, Flavourzyme liberates mainly polar amino acids, and not unpolar ones, as aimed at in the present study.
Intro:
“proteases” is no longer IUPAC standard should read peptidases
“polyphenol” should read polyphenols (amend abstract also)
M&M:
How was the identity of the Forastero variety confirmed?
Space between number and degree Celsius sign
DB-Wax (30 m x 250 m x 0.25 m) correct the units of this column
line 127 “for 0 minutes” ?
R&D
The names of amino acids should be preceded by a small capital (“L”) to indicated stereoforms
ILeucine should better read L-iso-Leucine
Table 4: Chemical names should start with a capital letter. Which Isopentanal? Amyl is now pentyl, butyrate is butanoate, propionate is now propanoate, caproate is hexanoate, same with the free acids, valerate is pentanoate etc, isodulcitol is not volatile enough for GC
analysis, 4,4-dimethyl-6-Heptyn-2-one is a very questionable identification - doublecheck
What was missed:
Authors have placed dried cocoa beans in buffer and incubated them for a couple of hours.Most of the analytical data show a minor effect only. Did they observe swelling? Did the enzyme penetrate the beans, or was the activity just working in the outer layer? Bromelain is
not a small molecule (I guess about 30 kDa). I could imagine that a preceding soaking would have facilitated the working of the enzyme and had given better results.
No sensory assessment was carried out. It is well known that any pre-treatment of a food may alter the flavour profile into undesirable directions because the genuine balance of compounds was changed. Some more data on the olfactory differences would be very much
appreciated.
Author Response
Response to Reviewer 3
[General comment] This is a well-written, straightforward, but somewhat premature paper on the use of a peptidase for the pre-treatment of non-fermented cocoa beans with the goal to improve the flavor profile.
Similar work, for example using the cheap peptidase mix Flavourzyme from Novo, has been done before. However, Flavourzyme liberates mainly polar amino acids, and not unpolar ones, as aimed at in the present study.
Response: We thank reviewers for appreciation our work. we have added a few sentences (Lines 259-260) to explain it as follows :
“During the fermentation of cocoa beans, the primary precursors that contribute to flavor development are hydrophobic amino acids formed during fermentation, particularly alanine, leucine, isoleucine, phenylalanine, and valine[36,37]”
“Bromelain has a broad specificity for protein breakdown both polar amino acids and hydrophobic amino acids [38]”. Cleavage by bromelain is mainly in hydrophobic amino acid residues, while non-polar amino acid residues remain at the C-terminus of the peptide. Its specific cleavage is at the arginine-alanine and alanine-glutamic acid bonds [14,17]
[Comment-1] “proteases” is no longer IUPAC standard should read peptidases “polyphenol” should read polyphenols (amend abstract also)
Response: we have revised “proteases” to peptidases “polyphenol” to “polyphenols”
[Comment-2] How was the identity of the Forastero variety confirmed?
Response: Cocoa beans of the Forastero variety were collected from cocoa farmers in Patuk, Gunung Kidul, Yogyakarta, Indonesia. Farmers in Patuk, Gunung Kidul plant cocoa beans of the Forastero variety with the characteristics of purple beans. Below we include pictures of the Forastero variety of cocoa pods and beans that were harvested and used in this research.
[Comment-3] Space between number and degree Celsius sign DB-Wax (30 m x 250 m x 0.25 m) correct the units of this column line 127 “for 0 minutes”?
Response: we have revised space between number and degree Celsius sign DB-Wax (30 m x 250 m x 0.25 m) to (30 m x 250 µm x 0.25 µm) as follows :
“The column used was DB-Wax (30 m x 250 µm x 0.25 µm), and the initial column temperature was 40 °C for 5 min then increased by 3°C/min to 220°C”
for 0 minutes means: a temperature of 220°C was not held.
[Comment-4] The names of amino acids should be preceded by a small capital (“L”) to indicated stereoforms ILeucine should better read L-iso-Leucine
Response: The names of amino acids have been revised to small capital. ILeucine has been revised to L-iso-Leucine.
[Comment-5] Table 4: Chemical names should start with a capital letter. Which Isopentanal? Amyl is now pentyl, butyrate is butanoate, propionate is now propanoate, caproate is hexanoate, same with the free acids, valerate is pentanoate etc, isodulcitol is not volatile enough for GC analysis, 4,4-dimethyl-6-Heptyn-2-one is a very questionable identification - doublecheck
Response: Isopentanal is the same as 2-methyl-Butanal, Isopentanal has been revised to 2-methyl-Butanal. Amyl has been revised to pentyl, butyrate has been revised to butanoate, propionate has been revised to propanoate, caproate has been revised to hexanoate, and valerate has been revised to pentanoate. 4,4-dimethyl-6-Heptyn-2-one4 has been removed from table 4.
[Comment-6] Authors have placed dried cocoa beans in buffer and incubated them for a couple of hours.Most of the analytical data show a minor effect only. Did they observe swelling? Did the enzyme penetrate the beans, or was the activity just working in the outer layer? Bromelain is not a small molecule (I guess about 30 kDa). I could imagine that a preceding soaking would have facilitated the working of the enzyme and had given better results.
Response: As a control, unfermented cocoa beans were incubated in an acetate buffer without the addition of bromelain. The increase in DH could be explained by the action of endogenous peptidases. However, the addition of bromelain as exogenous peptidases significantly increased the DH of cocoa beans much higher than incubating cocoa beans with acetic acid without adding bromelain. This indicates that the bromelain enzyme can penetrate into the beans and play a role in hydrolysis.
This is also supported by the analysis of the soluble protein content in the buffer-enzyme solution (Figure 2) was performed to determine if the enzyme protein could be diffused into unfermented dried cocoa beans during the hydrolysis process. This can be confirmed by comparing the buffer with the addition of the enzyme before hydrolysis with after hydrolysis. The results showed that the soluble protein was significantly reduced (p <0.05) after hydrolysis compared to before hydrolysis (Figure 2). The difference between the buffer with the addition of enzymes before and after hydrolysis proves that the enzyme can be diffused into the beans during the hydrolysis process and play a role in the hydrolysis of cocoa beans.
[Comment-7] No sensory assessment was carried out. It is well known that any pre-treatment of a food may alter the flavour profile into undesirable directions because the genuine balance of compounds was changed. Some more data on the olfactory differences would be very much appreciated.
Response: We thank the reviewer for this advice. In this research we focused on the development of flavor precursors and desired volatile compounds as a flavor of roasted cocoa beans, Therefore, sensory analysis can be used for further research.

Reviewer 4 Report
The authors looked at the impact of varying levels of bromelain had on amino acids, polyphenol content, and volatile compounds of cocoa beans.
Line 80 - 82: The method is confusing and needs to rewritten to improve clarity.
Why was an internal standard not utilized for your GC analysis? It's difficult to compare two different runs when no internal standard is used.
Line 182: You mention that bromelain increased enzyme hydrolysis to 76.98 - 83.07% for fermented cocoa beans. Not sure where this data came from since you only discuss it here and in the results.
Table 3: Are these values based upon mean+/-SD. Did you run statistics on this to see if there was true differences between the different treatments.
The GC data is very difficult to compare based when using area with no internal standard, because how are you sure that differences are truly as large or small as you indicate when you are not using the same volume in the samples.
Author Response
Response to Reviewer 4
[General comment] The authors looked at the impact of varying levels of bromelain had on amino acids, polyphenol content, and volatile compounds of cocoa beans.
Response: We thank the reviewers who have looked at the impact of varying levels of bromelain had on amino acids, polyphenol content, and volatile compounds of cocoa beans.
[Comment-1] Line 80 - 82: The method is confusing and needs to rewritten to improve clarity
Response: we have rewritten the method (Line 94-97) as follows:
“Crude bromelain at concentrations of 3.5, 7, and 10.5 U/mL was added to a 0.05 M acetate buffer solution pH 6. The solution was homogenized. Unfermented cocoa beans were added with a ratio of acetate buffer: cocoa beans 1:3 (w/ v) then incubated for 4, 6, and 8 hours at 50°C in a water bath shaker (Memmert WNB14, Germany)”
[Comment-2] Why was an internal standard not utilized for your GC analysis? It's difficult to compare two different runs when no internal standard is used.
Response: no internal standards have been used in the GC-MS analysis because the results were in percent area (%).
[Comment-3] Line 182: You mention that bromelain increased enzyme hydrolysis to 76.98 - 83.07% for fermented cocoa beans. Not sure where this data came from since you only discuss it here and in the results.
Response: The highest degree of hydrolysis with the addition of bromelain was at a concentration of 3.5 U/ml for 8 hours resulting in DH of 40.13% and 10.5 U/ml for 6 hours of 42.95%, compared to fermented cocoa beans resulting in DH of 50.70%. So, it can be concluded that the addition of exogenous protease (bromelain) significantly increased the DH of cocoa beans up to 76.98 – 83.07% of fermented cocoa beans.
[Comment-4] Table 3: Are these values based upon mean+/-SD. Did you run statistics on this to see if there was true differences between the different treatments.
Response: Yes. we have added the standard deviation values in Table 3. free amino acid of cocoa beans hydrolyzed by bromelain.
[Comment-5] The GC data is very difficult to compare based when using area with no internal standard, because how are you sure that differences are truly as large or small as you indicate when you are not using the same volume in the samples.
Response: no internal standards have been used in the GC-MS analysis because the results were in percent area (%).

Round 2
Reviewer 1 Report
The authors have extensively revised the manuscript. The Abstract section has been completed, the Introduction has been thoroughly revised, and relevant international literature has been added. The Material and Methods section has been clarified in terms of text, tables and figures. The PCA section has been added to the manuscript. The conclusion section was structured and made more logical. Overall, the manuscript has improved a lot, I recommend the publication of the manuscript.
Reviewer 4 Report
The improvements are acceptable.